# Technical efficiency of national HIV/AIDS spending in 78 countries between 2010 and 2018: A data envelopment analysis

**Kasim Allel** [1], **Gerard Joseph Abou Jaoude** [1]*, **Charles Birungi** [1,2], **Tom Palmer** [1], **Jolene Skordis** [1], **Hassan Haghparast-Bidgoli** [1]

**1** Institute for Global Health, University College London, London, United Kingdom, **2** United Nations Joint Programme on HIV and AIDS (UNAIDS), Harare, Zimbabwe

* gerard.jaoude.15@ucl.ac.uk

**Data Availability Statement:** Data are shared publicly at either the GBD (http://vizhub.healthdata.org/fgh/), WHO (apps.who.int/gho/data/), WB

## Abstract

HIV/AIDS remains a leading global cause of disease burden, especially in low- and middle-income countries (LMICs). In 2020, more than 80% of all people living with HIV (PLHIV) lived in LMICs. While progress has been made in extending coverage of HIV/AIDS services, only 66% of all PLHIV were virally suppressed at the end of 2020. In addition to more resources, the efficiency of spending is key to accelerating progress towards global 2030 targets for HIV/AIDs, including viral load suppression. This study aims to estimate the efficiency of HIV/AIDS spending across 78 countries. We employed a data envelopment analysis (DEA) and a truncated regression to estimate the technical efficiency of 78 countries, mostly low- and middle-income, in delivering HIV/AIDS services from 2010 to 2018. Publicly available data informed the model. We considered national HIV/AIDS spending as the DEA input, and prevention of mother to child transmission (PMTCT) and antiretroviral treatment (ART) as outputs. The model was adjusted by independent variables to account for country characteristics and investigate associations with technical efficiency. On average, there has been substantial improvement in technical efficiency over time. Spending was converted into outputs almost twice as efficiently in 2018 (81.8%; 95% CI = 77.64, 85.99) compared with 2010 (47.5%; 95% CI = 43.4, 51.6). Average technical efficiency was 66.9% between 2010 and 2018, in other words 33.1% more outputs could have been produced relative to existing levels for the same amount of spending. There is also some variation between WHO/UNAIDS regions. European and Eastern and Southern Africa regions converted spending into outputs most efficiently between 2010 and 2018. Rule of Law, Gross National Income, Human Development Index, HIV prevalence and out-of-pocket expenditures were all significantly associated with efficiency scores. The technical efficiency of HIV investments has improved over time. However, there remains scope to substantially increase HIV/AIDS spending efficiency and improve progress towards 2030 global targets for HIV/AIDS. Given that many of the most efficient countries did not meet 2020 global HIV targets, our study supports the WHO call for additional investment in HIV/AIDS prevention and control to meet the 2030 HIV/AIDS and eradication of the AIDS epidemic.

(https://data.worldbank.org/) or the UN (https://www.unaids.org/en/regionscountries/regions) websites/repositories.

**Funding:** The authors received no specific funding for this work.

**Competing interests:** The authors have declared that no competing interests exist.

## Introduction

HIV/AIDS remains a leading cause of global mortality and disease burden [1]. In 2020, 38 million people were living with HIV (PLHIV) across the world, 54.5% of which live in Eastern and Southern Africa followed by 15.3% in Asia and the Pacific region [2]. According to pre-COVID-19 projections, it was estimated that there will be less than 200,000 new infections and 400,000 deaths by 2030 [3]. Despite an overall reduction in the number of new HIV/AIDS infections and attributed mortality since 2010 (23% and 37%, respectively) [2], only 14 countries have met the global 2020 90-90-90 targets for diagnosis, treatment and viral suppression [4]. Substantial improvement in key HIV/AIDS indicators is therefore needed to meet the global target to end the HIV/AIDS epidemic by 2030 [5, 6]. In addition, existing challenges to improving national HIV/AIDS responses have been compounded by the COVID-19 pandemic. Interruptions in care for HIV/AIDS due to the COVID-19 pandemic is projected to substantially worsen treatment outcomes and set-back previous progress toward global 2030 target for viral load suppression [7].

Spending on HIV/AIDS in LMICs has increased from US$ 15.0 billion in 2010 to 21.5 billion in 2020. However, spending falls short of the target set by the UN General Assembly in the 2016 Political Declaration on Ending AIDS to invest at least 26 billion US dollars per year by 2020 [4, 8, 9]. Indeed, spending in 2020 was 29% lower than the US$ 26 billion global target and estimates suggest that additional efforts are required to meet the US$ 29 billion investment target for 2025 [8]. Additional resources are therefore likely needed to reach the global 95-95-95 target for 2030 (95% of PLHIV diagnosed, of whom 95% are treated, of whom 95% are virally suppressed). However, in light of a widening resource gap and the impact of the COVID-19 pandemic future funding, it is arguably more important than before to efficiently use existing resources to maximise impact. While this does not address the need to mobilise additional funding, it provides an additional lever for expanding fiscal and budgetary space for HIV responses.

Resources for healthcare are limited and must be used efficiently to maximise population health [10]. Allocative efficiency considers how to enhance social benefit, while technical efficiency focusses on the conversion of resources (inputs) to maximize a group of outputs [11–13]. A number of studies have investigated the allocative efficiency of HIV spending through different modelling approaches to inform priority setting and decision-making [14–18]. These include cost-effectiveness analyses and mathematical modelling techniques for resource optimization to meet HIV policy objectives. However, fewer studies have investigated the technical efficiency of HIV services or spending.

Data envelopment analysis (DEA) is widely used to estimate technical efficiency and measures the performance of Decision-Making Units (DMUs) by investigating their efficiency based on a given production set of inputs and outputs [26]. Using DEA, a recent study has measured the technical efficiency of the Indian Avahan HIV prevention project [19], whereas another in China investigated the efficiency of voluntary counselling and testing services [20]. Though, only two multi-country technical efficiency analyses have been published. One measured the technical efficiency of prevention of mother-to-child transmission (PMTCT) across 52 low- and middle-income countries (LMICs) in 2008 [21], while another study used DEA to estimate the technical efficiency of voluntary counselling, PMTCT and antiretroviral treatment (ART) across 68 LMICs for the years 2002–2007 [22]. All studies report low to moderate levels of efficiency, with inefficiencies of 50% estimated across countries [22].

The technical efficiency analysis of HIV spending has not been measured across countries for more recent years. However, this is needed given the limited evidence and substantial levels

of inefficiency previously reported. To address this gap, this study answers the following two questions:

1. What is the technical efficiency of national HIV spending in 78 countries from 2010 until 2018, and how does efficiency differ by region and income level?

2. What country-level independent variables are associated with technical efficiency across the 78 countries?

## Methods

### Data envelopment analysis

Data envelopment analysis (DEA), also known as frontier analysis, was first introduced by Charnes, Cooper and Rhodes in 1978 [23]. It is a performance measurement technique used for evaluating the relative efficiency of decision-making units (DMU's). It assesses how efficiently multiple inputs are able to produce multiple outputs through a maximisation process (maximising outputs from a given set of inputs) or minimisation process (minimising the inputs needed to produce a set of outputs) [24]. The model output is obtained from a non-parametric estimation for the optimum Pareto frontiers through which the efficiency of organisations can be determined. In other words, efficiency is a rank ordering of DMUs compared to a frontier of fully efficient DMUs. The radial distance of a DMU towards its frontier provides the measurement of its efficiency. Inefficient DMUs are enveloped by their efficient counterparts. DMUs can therefore improve their outputs without increasing inputs by maximising the ratio of the weighted sum of the outputs to the sum of the inputs. For the present study purposes, countries and their national HIV programmes are considered DMUs. They may be more efficient (higher efficiency scores) and closer to the estimated efficiency frontier or less efficient (lower efficiency scores) and further away from the frontier.

DEA analyses can use output- (maximisation) or input-oriented (minimisation) approaches to compute the efficiency frontier [19, 24]. We use an output-oriented approach because national HIV programmes aim to maximise outputs rather than minimise their inputs. Also, rapid structural changes on inputs (related to the level of spending on HIV by country) are unlikely to occur in the short-term [25, 26] and national HIV programmes arguably have less control over how many inputs they receive compared with the outputs they produce. We assumed variable returns to scale. This means an increase or decrease in any of our inputs or outputs is not translated into a proportional change in the outputs or inputs, respectively [27], given that national HIV programmes use multiple inputs to produce multiple outputs [28].

### Sample and main data sources

We collated data for an initial sample of 131 countries from publicly available sources. Input and output data were sourced from UNAIDS [29]. Data for independent variables were sourced from the World Bank (WB) [30], WHO Global Health Observatory [31], Global Burden of Disease (GBD) study by the Institute of Health Metrics and Evaluation [32] and the United Nations (UN). In total, 78 countries had information available on the main input and output variables (see Fig 1 for details and Table A of Section B in S1 Text). Countries were classified by WB income group [33] and WHO and UNAIDs regions [34, 35], as detailed elsewhere. Specifically, for WB income groups, low-income economies were defined as having a gross national income (GNI) per capita ≤$1,045 in 2020; lower middle-income economies a GNI per capita >$1,046 and ≤$4,095; upper middle-income economies a GNI per capita

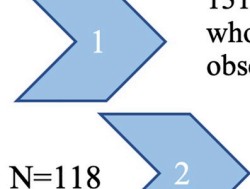

131 countries with data available on either PLHIV receiving ART or pregnant woman who received ARV for PMTCT since 2010 until 2018 with at least one data-point observed

N=118

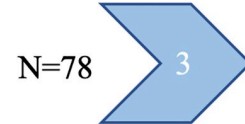

We dropped 13 countries without a single data-point available for PLHIV receiving ART throughout the time span

N=78

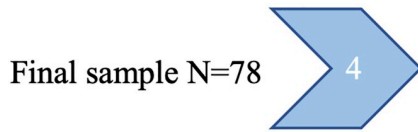

Pregnant women who received ARV for PMTCT added as a second output variable (Missing data= 33%; N=40)

Final sample N=78

Exogenous/control variables were imputed using a step-by-step protocol to complete missing data

**Fig 1. Sample schematic.** Note: ARV: Antiretroviral, ART: Antiretroviral therapy, PLHIV: People living with HIV, PMTCT: Prevention of Mother to Child Transmission.

>$4,096 and ≤$12,695; and high-income economies a GNI per capita ≥$12,696 or more. The final 78 countries included in the analysis represent every WHO and UNAIDS regions and WB income group (Table 1), with an average number of 7 data points for input and output variables between 2010 and 2018. This generated 581 observations in total. We used a different set of country-level characteristics as inputs, outputs, and independent variables for our main analysis following recent literature [19, 22, 36–38] that we describe below. Finally, some auxiliary variables were only used for a posterior sensitivity analysis.

## Input and output variables

Annual national HIV spending per person living with HIV (including new and existing cases) was used as an input [29]–which represents spending per prevalent case of HIV/AIDS. Data on total annual national HIV spending were extracted from UNAIDS for the years 2010 until 2018. Values are expressed in constant US$ 2019. Total annual HIV spending includes all spending, domestic and external, except for private and out-of-pocket spending. To generate the DEA input, annual national spending on HIV was divided by the annual number of PLHIV, or spending per person notified, to enable comparison across DMUs.

Two variables were used as annual outputs in the DEA model: (1) the percentage of PLHIV receiving Antiretroviral Therapy (ART) and (2) the percentage of pregnant women living with HIV receiving Antiretrovirals (ARV) for the prevention of mother-to-child transmission (PMTCT). We chose these variables because of data availability favouring cross-country comparability, and ART and PMTCT services accounted for more than 50% of total HIV spending (on average) in the countries with available information. The relationship between output and input variables was positive ($\rho = 0.17$–$0.27$) and satisfied the need for isotonicity.

## Independent variables

A series of independent variables were selected to investigate associations with technical efficiency. These variables were chosen based on the literature, interaction with HIV treatment,

**Table 1. Number of countries included by WHO region and World Bank income group (N = 78).**

| WHO region | Number of countries | (% out of the total)[a] |
|---|---|---|
| Africa Region | 36 | 78% |
| Eastern Mediterranean Region | 4 | 19% |
| European Region | 13 | 25% |
| Latin America and Caribbean Region | 16 | 46% |
| South-East Asia Region | 4 | 36% |
| Western Pacific Region | 5 | 19% |
| **World Bank income group** | **Number of countries** | |
| Low income | 21 | 78% |
| Lower middle income | 28 | 51% |
| Upper middle income | 26 | 47% |
| High income | 3 | 4% |
| **UNAIDS region** | **Number of countries** | |
| East and Southern Africa | 14 | 67% |
| West and Central Africa | 22 | 85% |
| North Africa and Middle East | 4 | 20% |
| Asia and Pacific | 9 | 24% |
| Eastern Europe and Central Asia | 10 | 59% |
| Latin America and the Caribbean | 16 | 49% |
| Western and Central Europe and North America | 3 | 8% |

Notes:

[a] This column refers to the percentage of the total countries in each region or income group.

and data availability. A total of 10 variables were used in our base model and four additional variables were included as auxiliary variables in the sensitivity analysis. The correlation between selected independent variables (using Pearson's coefficient) was investigated to avoid multicollinearity (see Table B of Section B in S1 Text). Our selected variables were not highly correlated between them (medium and low correlation levels encountered, Pearson coefficient <0.7). Multicollinearity tests using the variance inflator factor (VIF) for the adjusted truncated models were employed. We did not find high multicollinearity (mean VIF = 3.28, Table F of Section B in S1 Text) for our adjusted truncated regression. Details on the variables used, justification for inclusion and expected direction of their relationship with efficiency are included in Table 2. Additional variables were tested including a wide range of countries' characteristics (see S1 Data); however, most variables were removed due to collinearity, high correlation or null evidence provided within the literature.

## Missing data

We used a two-step procedure to address missing values from collated data (15% of the independent variables were missing; density of health posts, number of nurses, and HIV prevalence accounted for most). First, we imputed missing data points between years by using the average of the observable variables, following which we carried back or forward values from the earliest or latest observable data-points respectively. Third, for countries without data on a specific independent variable, we employed missing value imputation using multivariate normal regression methods, in which 50 imputations were performed, and then the final independent variable was generated using an averaged value (details on the model used and imputation diagnostics are found in Section A in S1 Text).

Table 2. Expected effect of independent variables on technical efficiency and justification for inclusion in the analysis.

| Independent variable | Definition and Justification [Expected effect for efficiency*] | Source |
|---|---|---|
| Rule of Law | Rule of law is a continuous variable ranging from -2.5 to 2.5 which indicates how well a country does in regard to their police force and crime rates, quality of contract enforcement, property rights, violence rates, among others. This has been previously included by Zeng and co-authors in their technical efficiency analysis of HIV spending [22]. Rule of law was included amongst a different set of government indicators having the highest association with the outcomes analysed. [+] | WB Database [30] |
| Antenatal Care Coverage (ACC), at least 4 visits (%) | It is an indicator of access and use of health care by pregnant women during their pregnancy. Antenatal care is crucial for pregnant women and also their infants. Receiving care increases the probability of receiving adequate and effective interventions to improve their health, including PMTCT which is one of the outputs in the main model. Countries with higher rates of antenatal care coverage may also have better quality of health services and more equitable distribution of health care. Specifically, ACC is a vital component to prevent mother-to-child transmission of HIV as it increases the opportunity to reach HIV-positive pregnant women through the integration of HIV testing into antenatal care routine. [+] | WHO Global Health Observatory [31] |
| Gross National Income (GNI) per capita in 2019 PPP, in current USD | Continuous variable. GNI indicates the average before-tax income per citizen of a country during a year. Higher income can be associated with higher wealth, infrastructure and living standards, all of which may result greater efficiency scores. [+] | WB Database [30] |
| Human Development Index (HDI) | HDI is an index ranging from 0 to 100 indicating how developed a country is in terms of education, income per capita and life expectancy. Higher HDI scores indicate that countries perform better in the three dimensions. This may be associated with greater HIV efficiency scores given the dimensions of health and life expectancy are capture by the HDI [+] | WB Database [30] |
| Population per squared kilometre | Continuous variable. A higher number of people per $km^2$ (density) suggests higher rates of urbanization, as well as lower rates of rurality, and easier access to healthcare facilities. [+] | WB Database [30] |
| Current health expenditure per capita, PPP (current international in 2019 USD) | Continuous variable. Greater levels of per capita expenditure on health are highly associated with improved access to care as health systems can act efficiently in handling different diseases. Higher expenditures indicate more technology and infrastructure in hospitals and the health system as a whole. [+] | WB Database [30] |
| Current health expenditure as a % of Gross Domestic Product (GDP) | The proportion of the GDP accounted for by health spending can be considered a proxy for government commitments to health. Governments with higher investments on health may have better structured health systems and so achieving greater efficiency levels. [+] | WB Database [30] |
| Prevalence of HIV/AIDS per 100,000 people | Greater values of HIV/AIDS prevalence may be associated with weaker and more fragmented health systems which cannot control and treat the disease in an efficient manner. [−] | WHO Global Health Observatory [31] |
| Proportion of Total HIV Spending from Out-of-Pocket (OOP). Sources | A greater proportion of HIV/AIDS spending comprised of OOP signifies higher financial barriers for PLHIV to access care, reduced coverage, is therefore linked to greater inequality and a lack of efficiency. [−] | IHME Financing Global Health [39] |
| Proportion of Total HIV Spending from Development Assistance for Health Spending (DAHS) | Development assistance and financial aid from external funding for HIV may enhance the efficiency of the health systems through the prevention of treatable health conditions, investment on hospital infrastructure, more technical support and appropriate staff, among other improvements in terms of drugs, medicines, and treatments. Even though higher aid may be associated with clear enhancements in healthcare, it might be the case that higher levels of aid is also a reflection of dependence on external money due to a lack of infrastructure and health system resources which can lead to greater inefficiencies. [+ or -] | IHME Financing Global Health [39] |
| Government spending per total HIV spending ratio | Ratio indicating the proportion between public spending and HIV-specific expenditures. Government public spending includes subsidies, property income, compensations of employees, education, and social benefits. If the overall ratio is substantially high, then the country has other priorities than HIV or simply the HIV prevalence is too small. Efficiency will therefore depend on contextual variation. [+ or -] | IHME Financing Global Health [39] |

(*Continued*)

**Table 2.** (Continued)

| Independent variable | Definition and Justification [Expected effect for efficiency*] | Source |
|---|---|---|
| External health expenditure as a % of Current Health Expenditure [CHE] | Higher external health spending as a proportion of total CHE indicates a higher reliance on external funding which can reflect lower overall capacity to provide healthcare leading to greater inefficiency. [–] | World Bank Open Database [30] |
| Number of nurses per 10,000 people | Continuous variable. A greater number of nurses has been associated with better structured and high-performing health systems, as the healthcare service is not overwhelmed due to a lack of staff. Even though increases in the number of nurses may favour better access to healthcare and reflect higher health investments, it might be that this relationship is not always the most efficient and may cause additional costs to incur. [+ or -] | WHO Global Health Observatory [31] |
| Total density of health posts per 100,000 people | Continuous variable. A higher number of health posts may indicate greater investments in health infrastructure and widened access to health services for people across a country [+] | WHO Global Health Observatory [31] |

Notes:

* The expected effect is for technical efficiency, but our models compute the reciprocal of technical efficiency (i.e. inefficiency) for statistical purposes. Therefore, the expected effects are the opposite when evaluating inefficiency.

## Statistical analysis

The two-stage double-bootstrap DEA approach, developed by Simar-Wilson [40, 41], is used in this paper to explore how different country-level independent variables are associated with higher or lower efficiency scores. The Simar-Wilson double-bootstrap approach is increasingly being used to investigate technical efficiency in the health sector [19, 28, 37, 38]. In the first stage, this linear programming approach adjusts initial efficiency scores by the potential biases caused by other independent variables. In the second stage, the scores are bias corrected from the previous step and then used in a truncated regression model that controls for independent variables which may affect outputs. This approach attempts to improve the estimated technical efficiency scores by eliminating potential biases caused by existing measurement errors or serial correlations [19]. Full descriptions of the DEA algorithms used can be found elsewhere [37, 40, 41] and in Section A in S1 Text.

We estimated bias-corrected efficiency scores using an output-oriented DEA model with variable returns to scale and adjusted to independent variables [40, 41]. We used 1,000 bootstrap replications in the first loop of Simar and Wilson's (2007) approach and 3,000 bootstrap replications in the second loop for bias-correction of technical efficiency scores. 95% CIs computations were built based on the distance function, i.e., the reciprocal of efficiency score, ranging between one to infinity. Then, we computed the reciprocal of this value (inverse), to generate bias-corrected efficiency scores between 0 (least efficient) and 1 (most efficient). All analyses were carried out in RStudio version 3.3 using the rDEA package (dea.env.robust command, available at https://github.com/jaak-s/rDEA). As per the software programme, truncated multivariate regression was employed using the reciprocal of the efficiency scores as dependent variable. Therefore, the direction and size of the estimates reflect their impact on the inverse of the efficiency scores (i.e., inefficiency).

## Sensitivity analysis

To investigate the robustness of our main model technical inefficiency scores (reciprocal of efficiency), we carried out two sensitivity analyses. In the first analysis, we tested three different models by including auxiliary variables that have been relevant in the literature [22, 36, 38] but contained substantial missing data that had to be imputed. First, we added the number of

nurses and health posts to the base model (A). Second, we incorporated the percentage of HIV spending from government sources and the percentage of current health expenditure accounted for by external sources (development assistance for health spending) to the base model (B). Third, we combined models A and B into a single model. In the second analysis, we removed the lowest and highest 5% outliers of our outputs and inputs (separately and jointly) given that the DEA method is highly sensitive to outliers [42].

# Results

## Descriptive statistics

Table 3 presents the descriptive statistics of our sample. Average annual HIV spending is around US$488 per person between 2010 and 2018 (SD = 506). Our output variables indicate that 36.8% of the PLHIV are receiving ART treatment (SD = 15.3), whereas 64.7% of the total pregnant women in need of ARV for PMTCT are receiving ARV (SD = 25.4). The number of PLHIV receiving ART varies considerably between countries, while the percentage of pregnant women receiving ARV was highest in Southern African and South Asian countries (Fig 2). HIV prevalence is 2.56% across the countries (SD = 5.23) with the highest levels observed in Africa, specifically in Southern, Central, and Western Africa (Fig 2).

**Table 3. Descriptive statistics and imputed missing data final sample (N = 78 countries).**

| Name of the variable | Mean | SD | P25th | P75th | Min | Max |
|---|---|---|---|---|---|---|
| *Output variables* | | | | | | |
| PLHIV receiving ART (%) | 36.82 | 15.32 | 25.06 | 48.37 | 9.76 | 71.83 |
| Pregnant women receiving ARV for PMTCT (%) | 64.69 | 25.41 | 46.00 | 87.00 | 4.00 | 100.00 |
| *Input variable* | | | | | | |
| Total HIV spending per person in USD | 488.25 | 505.53 | 201.51 | 601.51 | 45.12 | 3,501.51 |
| *Independent variables* | | | | | | |
| Rule of Law | -0.46 | 0.73 | -0.96 | -0.05 | -2.42 | 1.93 |
| Antenatal Care Coverage (%) | 70.69 | 19.75 | 55.50 | 89.30 | 19.10 | 100.00 |
| GNI pp | 3,766.78 | 4,265.35 | 775.71 | 5,840.05 | 100.00 | 33,994.41 |
| CHE as %GDP (%) | 5.78 | 2.20 | 4.43 | 6.73 | 1.35 | 20.41 |
| CHE pp in USD | 207.72 | 270.31 | 39.55 | 278.53 | 1.00 | 2421.69 |
| Population per KM$^2$ | 96.98 | 103.77 | 24.84 | 121.27 | 3.39 | 666.61 |
| HDI | 59.26 | 13.90 | 46.7 | 72.2 | 32.60 | 84.70 |
| HIV prevalence (%) | 2.56 | 5.22 | 0.30 | 1.60 | 0.01 | 28.20 |
| Out-of-pocket expenditure as % of the total HIV spending | 0.06 | 0.08 | 0.01 | 0.07 | 0.00 | 0.40 |
| DAHS per total HIV spending ratio | 0.50 | 0.31 | 0.20 | 0.78 | 0.00 | 0.98 |
| *Auxiliary variables* | | | | | | |
| Government spending per total HIV spending ratio | 0.42 | 0.30 | 0.15 | 0.68 | 0.02 | 1.18 |
| External expenditure as % of CHE | 14.52 | 16.44 | 0.83 | 23.43 | 0.00 | 86.20 |
| Number of nurses per10,000 people | 21.28 | 25.53 | 5.70 | 23.06 | 0.61 | 122.73 |
| Total density of health posts per 100,000 people | 10.97 | 17.65 | 0.10 | 15.14 | 0.10 | 272.64 |

*Notes*: pp: per capita. ART: antiretroviral therapy. ARV: antiretroviral. GNI: Gross National Income. CHE: current health expenditure. HDI: human development index. PMTCT: prevention of mother-to-child transmission. DAHS: Development assistance for health spending. Fig D of Section B in S1 Text exhibits the distribution of our input and output variables.

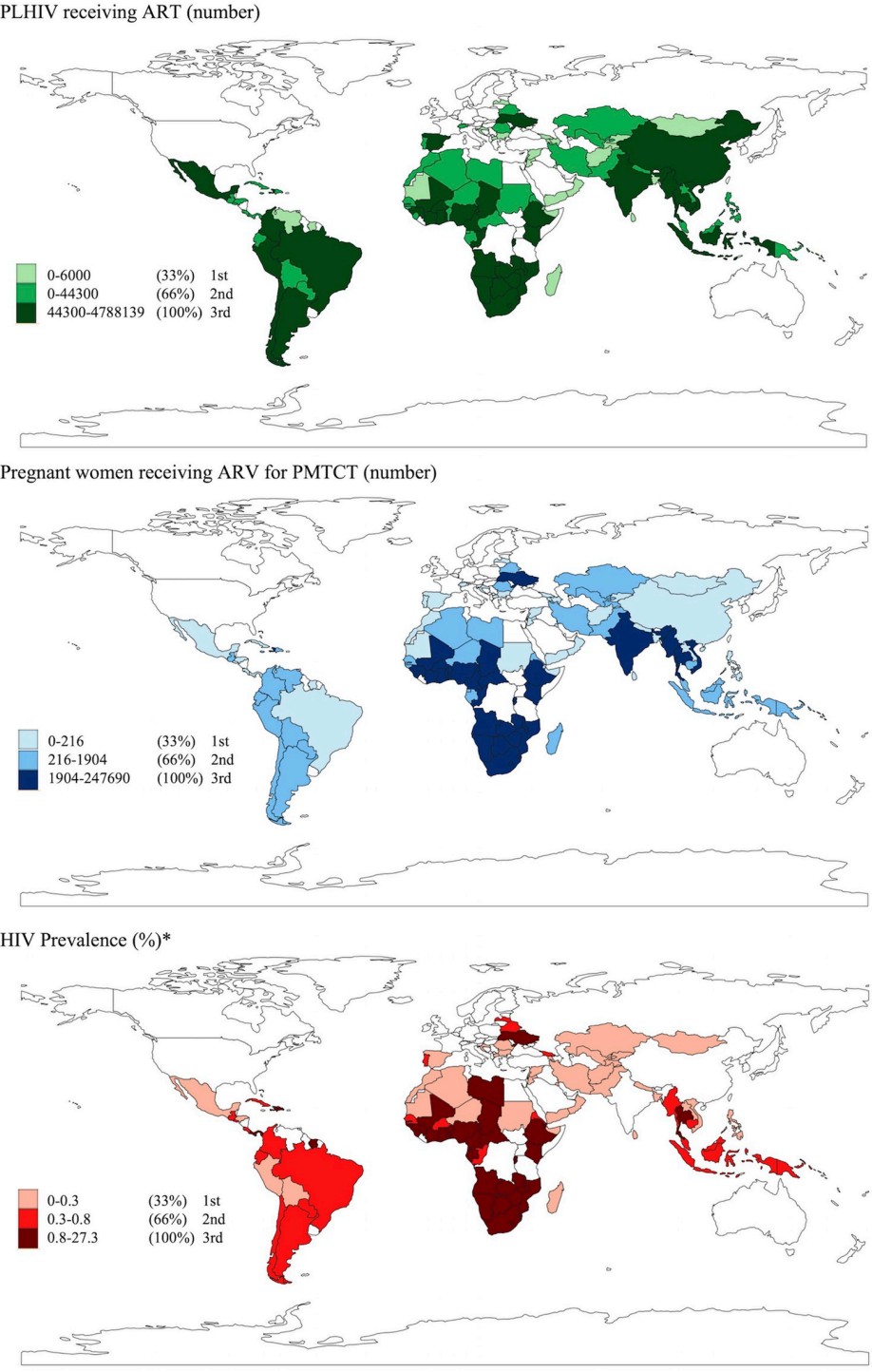

**Fig 2. Raw input and output variables terciles by country (N = 78). Notes**: Raw variables presented before transforming them into proportions. * White areas for HIV prevalence were missing but some of them were imputed afterwards for analytical purposes. Data used to generate the maps is included in S1 and S2 Data. Maps were created using "QGIS Geographic Information System" from the Open Source Geospatial Foundation Project (http://qgis. osgeo.org). The QGIS Geographic Information System is a free and open source software, for more details on copyright: https://www.qgis.org/en/site/getinvolved/governance/trademark/index.html#:~:text=QGIS%20trademarks %2C%20service%20marks%2C%20logos,all%20uses%20of%20QGIS%20Marks.

## Technical efficiency of national HIV spending

The average technical efficiency of HIV spending, measured by bias-corrected efficiency scores, has increased from 47.5% in 2010 (95% CI = 43.4, 51.6) to 81.8% (95% CI = 77.6, 86.0) in 2018 (Fig 3). In other words, while 52.5% more outputs could have been produced in 2010 for the amount of spending, in 2018 an additional 18.2% more outputs could have been produced for the same amount of spending. Constant improvement in technical efficiency is observed between 2010–2018 across countries, with an average annual improvement of 5 percentage points (pp). The average bias-corrected efficiency score across country years was 67.0%, with a 1.6% (average bias calculated = -0.011) correction of the initial computed efficiency scores. Initial DEA, and bias-corrected DEA efficiency scores by country and year are presented in Table C of Section B in S1 Text. The distributions of efficiency scores by country and across countries are displayed in Figs B and C of Section B in S1 Text. The correlation coefficient (Pearson's) between initial and bias-corrected efficiency scores was $\rho$ = 0.996.

As shown in Fig 3, technical efficiency scores increased most between 2010 and 2015, following which there is a more gradual increase from 2015 to 2018. However, there is a substantial spread of efficiency scores between 2010 and 2013. From 2014 onwards, there is a narrowed range for efficiency scores (concentrated around the mean)–until technical efficiency reached its highest level in 2018. High income countries are the most efficient between income groups, with the narrowest 95% CIs and a large concentration of data around their median (Fig 4). High-income countries performed better than low-income counterparts (efficiency scores of 97% and 60%, respectively), followed by upper and lower middle-income countries which had median efficiency levels of 77% and 63%, respectively. Across WHO regions, the Eastern Mediterranean region (EMRO) is the least efficient (33%), whereas the European region is the most efficient (83%). Covering UNAIDS regions, we can see that Eastern and Southern Africa (78%) had one of the highest efficiency levels following Western and Central European and North American countries (99%). However, results by income groups and regions vary over time, as seen in Fig 5, and Fig E of Section B in S1 Text. The largest change in technical efficiency over time is observed for Eastern Mediterranean countries, which has been the region presenting most inefficient countries over time, but their efficiency levels have increased by 48% between 2010 and 2018. Even though European countries outperform other countries, the Western Pacific region exhibited the second highest efficiency scores in the most recent year (2018). Similarly, while high-income countries outperformed other income groups over time, low-income countries surpassed lower-middle income countries, becoming more efficient from 2016 onwards. Overall, sampled countries could improve their efficiency by up to 20% (approximately) in 2018, although room for improvement varies by income and region group. Top performers by income group are Chile, Spain and Portugal (high income), Cuba, Dominican Republic, Romania and Suriname (upper-middle income), Bolivia, Cambodia and Cameroon (lower-middle income) and Benin, Gambia, Mozambique and Rwanda (low income).

While differences between income groups and regions are noteworthy, the technical efficiency of HIV spending differs most between countries from 2010 to 2018 (Fig 6). Efficiency scores range between 22% and 98%, a difference of 76pp between the least efficient (Indonesia and Sudan) and most efficient (Romania) countries. The average technical efficiency across country-years was 67% over time, and most countries (90% of observations) were between 26% and 98% efficient. Most of our sampled countries (83%) reporting the highest HIV-burden (adults' prevalence above 5%) lied above the average technical efficiency across country-years (e.g. Malawi, Mozambique, Lesotho, Zimbabwe, Zambia) except for Equatorial Guinea.

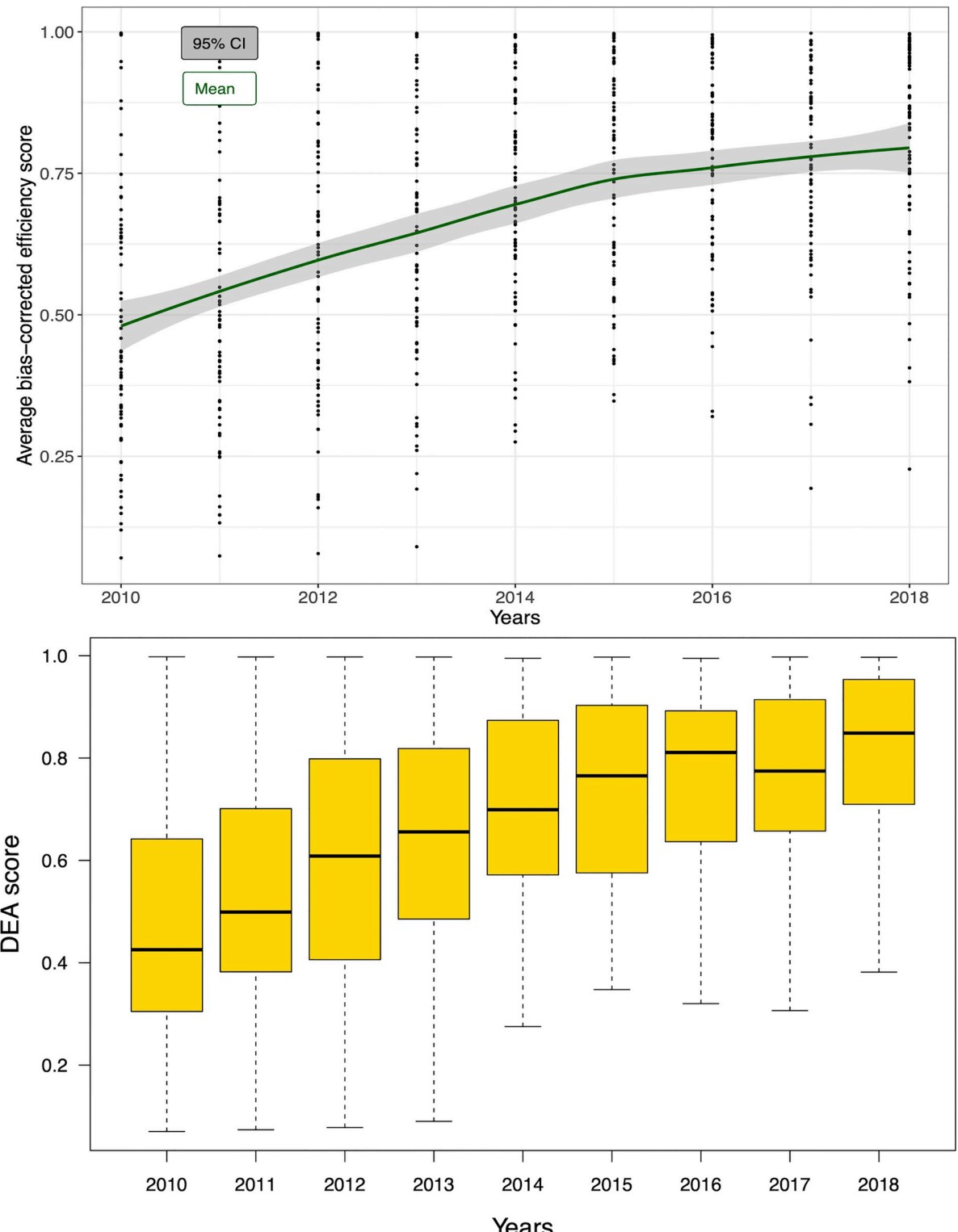

**Fig 3. Mean and 95% CI of our adjusted technical efficiency scores over the years.** *Notes*: black dots stand represent observations per year. Bottom figure presents the robust reciprocal of bias-corrected efficiency scores. Horizontal lines represent median values, boxes show the IQR, whiskers show data points within 1.5x|IQR|. DEA scores stand for efficiency scores.

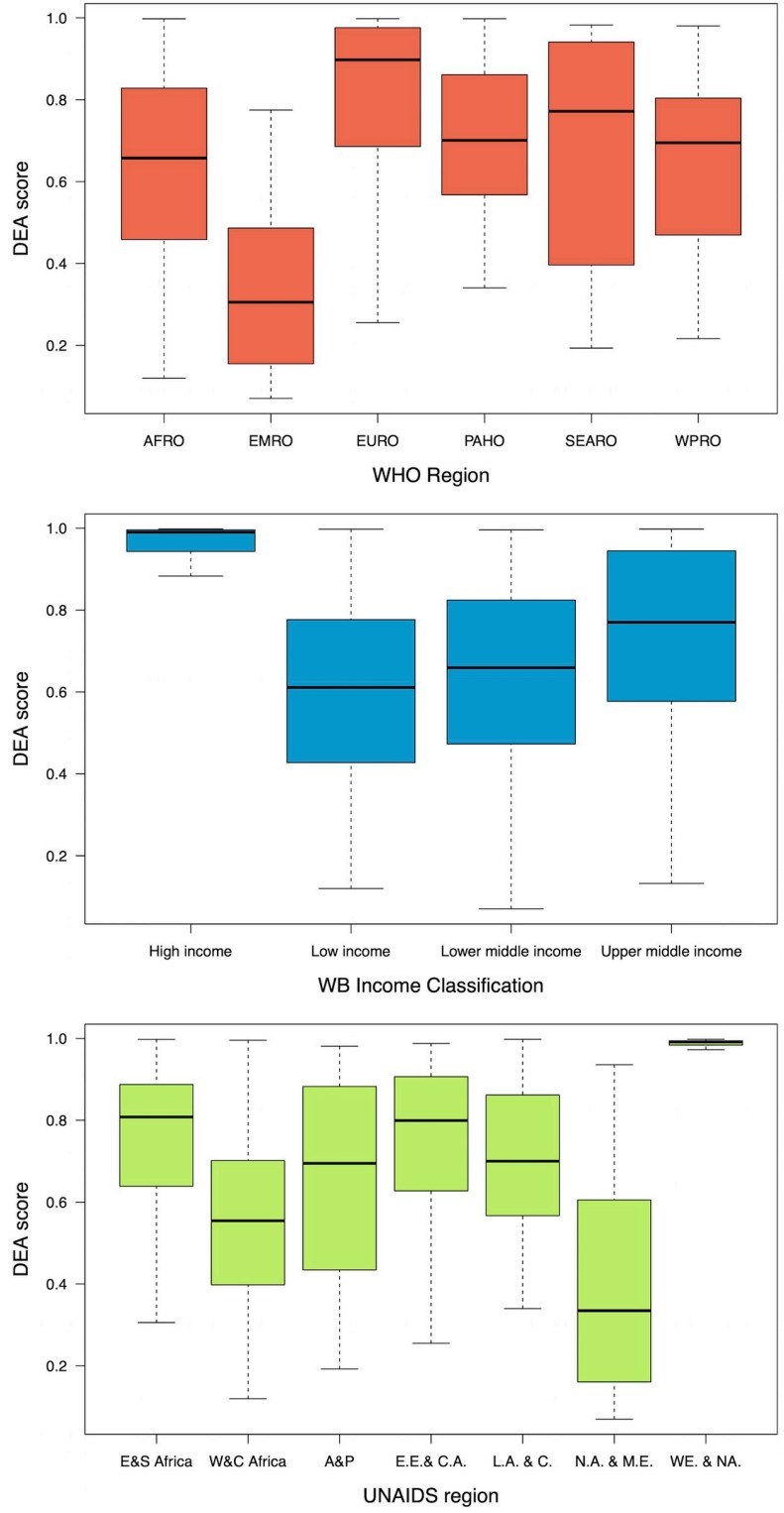

**Fig 4. Bias-corrected efficiency scores from the base model by year, WB income group and WHO region.** *Notes*:
Robust reciprocal of bias-corrected efficiency scores were used. Horizontal lines represent median values, boxes show
the IQR, whiskers show data points within 1.5x|IQR|. E&S stands for Easter and Southern, W&C for Western and
Central, A&P is Asia and the Pacific, E.E. & C.A. is Eastern Europe and Central Asia, L.A.&C.A. Latin America and the
Caribbean, N.A.&M.E. is North Africa and Middle East, and WE.&NA. is Western and Central Europe and North
America.

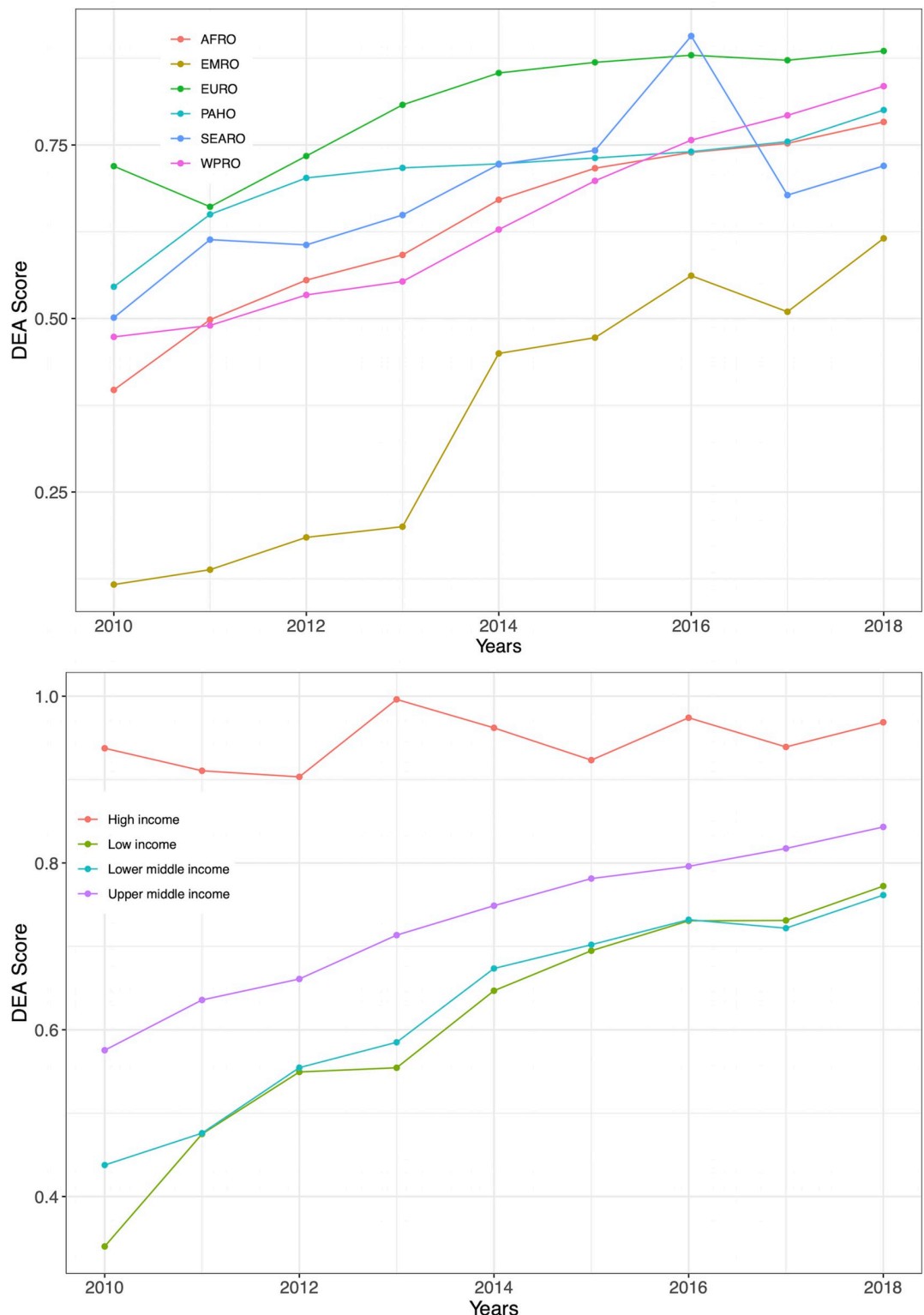

**Fig 5. Average bias-corrected efficiency scores by WHO region and WB income group.** *Notes*: Values extracted from the base model. Same graph by UNAIDS region is shown in Fig E of Section B in S1 Text. DEA scores stand for efficiency scores.

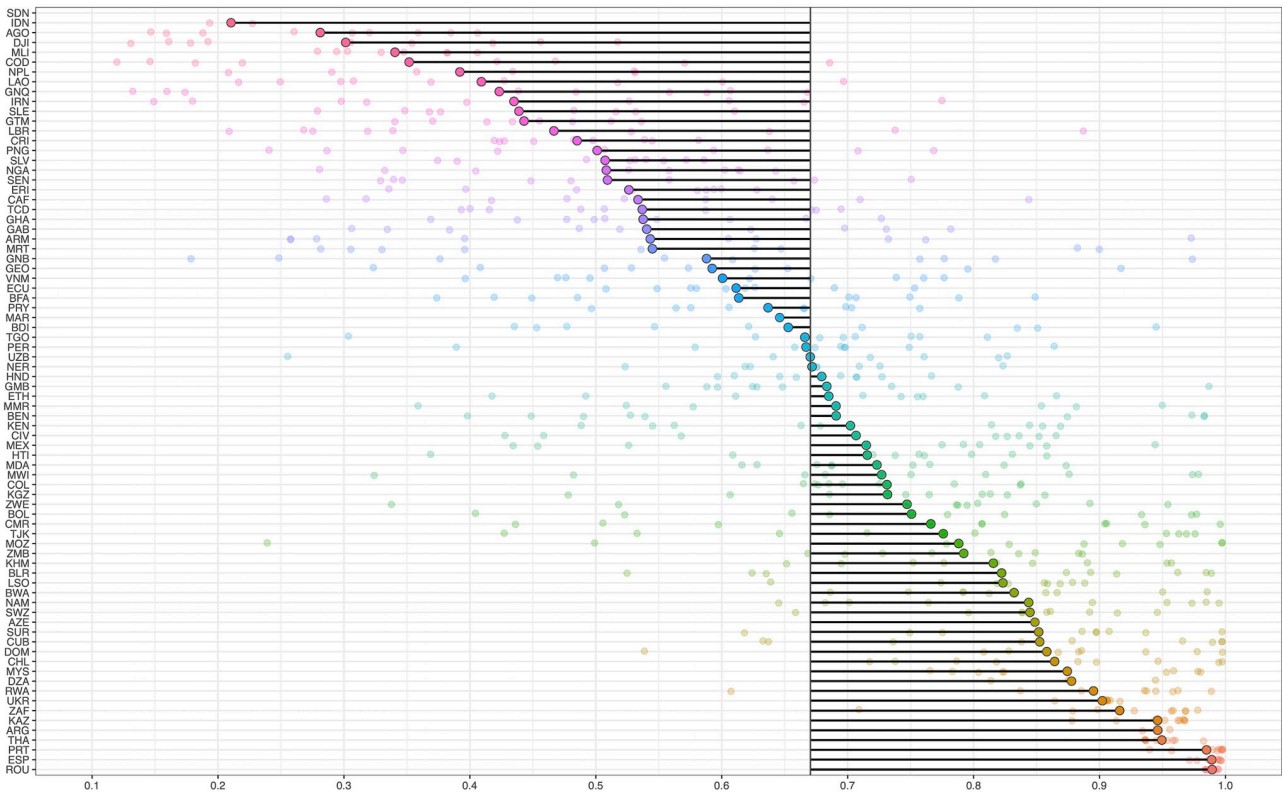

**Fig 6. Average bias-corrected efficiency scores by country (N = 78).** *Notes*: Average technical efficiency across country-years is depicted by the middle vertical line. Dots present each observation per country. Y-axis shows country codes. SDN has below 0.1 values (see Table C of Section B in S1 Text for further details on scores estimated per country).

Initial efficiency scores, bias estimates and bias-corrected efficiency scores are included in full by country-year in Table C of Section B in S1 Text).

## Independent variables associated with technical inefficiency of national HIV spending

Eight of the ten independent variables investigated are significantly associated with average bias-corrected efficiency scores in the truncated multivariate regression (Table 4). Three variables are negatively associated with average inefficiency (but positively with efficiency). In decreasing order of coefficient size these are: Rule of Law (Coeff = -15.49, p-value<0.001), HIV prevalence (Coeff = -3.17, p-value<0.001), and HDI (Coeff = -1.55, p-value<0.001). These variables have an inverse association with technical inefficiency, in other words a one-unit increase in these variables is associated with an increase in average efficiency score (decrease for average inefficiency). For example, a 1% increase in HIV prevalence is associated with a 3.2% decrease (increase) in average inefficiency (efficiency).

In contrast, five variables were positively associated with technical inefficiency. These are, in decreasing order of their coefficient size: GNI per capita (Coeff = 9.31, p-value<0.001), CHE per capita (Coeff = 7.14, p-value = 0.001), out-of-pocket expenditures as percentage of total HIV spending (Coeff = 0.60, p-value<0.001), DAHS as a percentage of total HIV spending (Coeff = 0.13, p-value = 0.026), and population density (Coeff = 0.02, p-value = 0.046). A rise in one of these variables is associated with an increase (decrease) in technical inefficiency

**Table 4. Results of the robust coefficients in the truncated regression for the reciprocal of the efficiency scores (i.e. inefficiency) (N = 78).**

| Variables | Coeff. | SE | P-value |
|---|---|---|---|
| (Intercept) | -52.80 | 17.19 | 0.002 |
| Rule of Law | -15.49 | 4.15 | 0.000 |
| Antenatal Care Coverage | -0.08 | 0.09 | 0.396 |
| GNI per capita in USD | 9.31 | 2.54 | 0.000 |
| CHE as % of GDP | -1.36 | 0.80 | 0.088 |
| CHE per capita in USD | 7.14 | 2.09 | 0.001 |
| Population per KM$^2$ | 0.02 | 0.01 | 0.046 |
| HDI | -1.55 | 0.44 | 0.000 |
| HIV prevalence | -3.17 | 0.90 | 0.000 |
| OOP spending as % of the total HIV spending | 0.60 | 0.16 | 0.000 |
| DAHS per total HIV spending ratio | 0.13 | 0.06 | 0.026 |
| Sigma | 5.18 | | |

Notes: Base model without any outlier for spending in HIV/AIDs and outcomes (ART) + PMTCT. GNI: Gross National Income, CHE: Current Health Expenditure, GDP: Gross Domestic Product, HDI: Human Development Index. OOP: Out-Of-Pocket. DAHS: Development assistance for health spending. Coeff.: coefficients, SE: standard errors.

(efficiency). For example, a 1% increase in OOP or DAHS as a percentage of total HIV spending is associated with a respective 0.60% and 0.13% increase (decrease) in average technical inefficiency (efficiency).

Pearson's correlation and a visual relationship between all our independent variables and technical efficiency can be found in Table D and Fig A of Section B in S1 Text, respectively. The univariate (unadjusted) truncated models were consistent with our main adjusted truncated results (Table E of Section B in S1 Text). The direction of our estimates, which include imputed missing data, was similar with those using only non-imputed data (Table G of Section B in S1 Text).

## Sensitivity analysis

For the first sensitivity analysis (a), three alternative models were used to test the base model bias-corrected efficiency scores. Overall, scores varied by less than 1% compared with base model estimates (Fig 7). Model A in Table 5, which includes the number of nurses and health posts, is similar to the base model but finds a significant negative association between the number of nurses and average inefficiency (Coeff = -0.55, p-value<0.001), i.e. positive association with average efficiency. Model B finds that the two-additional sources of spending included, i.e. government spending as a percentage of total HIV and external expenditures as a percentage of CHE, are not significant predictors of technical inefficiency. However, average efficiency scores between models are almost identical. Last, the third model tested combined the independent variables separately added in Models A and B. The third model finds no other significant association in the additional independent variables with average efficiency, aside from number of nurses and density of health posts (Fig 7 and Table 5).

For the second analysis (b), three models were tested excluding the 5% upper, lower and both upper and lower outliers in our main model (Table H of Section B in S1 Text). The lower and upper 5%, outliers for our inputs and output, excluded from the whole sample, did not influence the reciprocal of the efficiency scores (inefficiency). On average, our main model

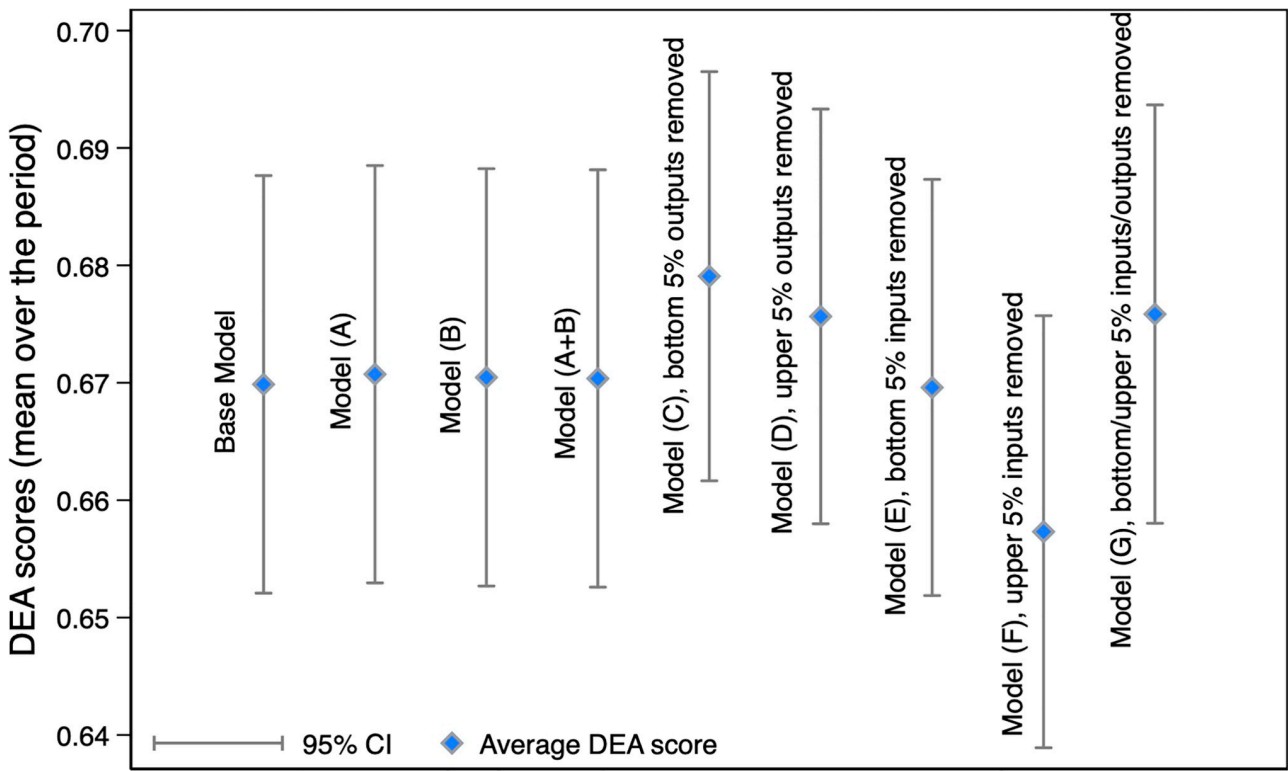

**Fig 7. Average bias-corrected efficiency scores by sensitivity analysis model.** *Notes*: 95% CIs were not added as y-axis scale is small. No notable differences are observed in average bias-corrected efficiency scores. T-test between the values were applied and p-value>0.1 for all comparisons. There were 76, 77, 77, 74, 72 number of countries and 643, 619, 659, 617, and 571 observations included for Model C, D, E, F, and G, respectively. DEA scores stand for technical efficiency scores.

scores varied by one percentage point after excluding either the upper or the lower outliers. The variation was not significant after removing all the outliers from the base model (Model G, Fig 7). Furthermore, there were no differences in the average efficiency score after comparing each model with the base model (t-test p-value>0.1), (Table H of Section B in S1 Text, panel B).

## Discussion

This study provides an updated estimate of the technical efficiency of HIV spending, between 2010 and 2018, in 78 countries accounting for approximately 50% of the global HIV burden. Country-level factors associated with average efficiency are also investigated. We used a double-bootstrap truncated regression DEA approach, which has not yet been applied to investigate the technical efficiency of HIV spending across countries.

Our findings showed that the global technical efficiency of HIV spending was 81.8% in 2018. In other words, 18.2% more outputs could have been produced globally for the same amount of spending. However, variation was observed in average efficiency between WHO regions and WB income groups, with higher income countries and the EURO, SEARO and WPRO performing especially well. Although global and regional efficiency improved substantially over time, by 34.3 percentage point since 2010, there remains scope to further reduce inefficiency. For example, 36.5% more outputs could have been produced for the same level of spending in the EMRO region in 2018. Even larger differences are observed between countries,

**Table 5. Results of the sensitivity analyses for the reciprocal of the efficiency scores (i.e. inefficiency) (N = 78).**

| A) Exploring by number of nurses and density of health posts | | | |
|---|---|---|---|
| Variables | Coeff. | SE | P-value |
| (Intercept) | -60.10 | 17.36 | 0.001 |
| Rule of Law | -14.66 | 3.79 | 0.000 |
| Antenatal Care Coverage | 0.04 | 0.08 | 0.622 |
| GNI per capita in USD | 8.60 | 2.28 | 0.000 |
| CHE as % of GDP | -1.43 | 0.71 | 0.043 |
| CHE per capita in USD | 5.18 | 1.60 | 0.001 |
| Population per $KM^2$ | 0.02 | 0.01 | 0.012 |
| Human Development Index (HDI) | -1.11 | 0.32 | 0.001 |
| HIV prevalence | -3.03 | 0.85 | 0.000 |
| OOP spending as % of the total HIV spending | 0.45 | 0.13 | 0.001 |
| DAHS per total HIV spending ratio | 0.16 | 0.05 | 0.003 |
| Number of nurses per 10K people | -0.55 | 0.16 | 0.001 |
| Density of health post per 100K people | 0.09 | 0.06 | 0.100 |
| Sigma | 4.84 | | |
| **B) Exploring by GS per total HIV spending and external expenditure as % of CHE** | | | |
| Variables | Coeff. | SE | P-value |
| (Intercept) | -55.13 | 19.60 | 0.005 |
| Rule of Law | -15.31 | 4.55 | 0.001 |
| Antenatal Care Coverage | -0.06 | 0.09 | 0.537 |
| GNI per capita in USD | 9.87 | 2.89 | 0.001 |
| CHE as % of GDP | -1.33 | 0.81 | 0.101 |
| CHE per capita in USD | 7.09 | 2.32 | 0.002 |
| Population per $KM^2$ | 0.02 | 0.01 | 0.057 |
| Human Development Index (HDI) | -1.66 | 0.51 | 0.001 |
| HIV prevalence | -2.75 | 0.87 | 0.002 |
| OOP spending as % of the total HIV spending | 0.74 | 0.22 | 0.001 |
| DAHS per total HIV spending ratio | 0.18 | 0.07 | 0.011 |
| GS per total HIV spending ratio | -0.70 | 0.47 | 0.139 |
| External expenditure as % of CHE | -0.13 | 0.09 | 0.153 |
| Sigma | 5.27 | | |
| **A+B) Exploring by number of nurses, density health posts, GS per total HIV spending and external expenditure as % of CHE** | | | |
| Variables | Coeff. | SE | P-value |
| (Intercept) | -74.57 | 24.44 | 0.002 |
| Rule of Law | -14.85 | 4.06 | 0.000 |
| Antenatal Care Coverage | 0.04 | 0.08 | 0.631 |
| GNI per capita in USD | 10.24 | 3.15 | 0.001 |
| CHE as % of GDP | -1.20 | 0.67 | 0.074 |
| CHE per capita in USD | 5.27 | 1.79 | 0.003 |
| Population per $KM^2$ | 0.02 | 0.01 | 0.015 |
| Human Development Index (HDI) | -1.18 | 0.38 | 0.002 |
| HIV prevalence | -2.79 | 0.89 | 0.002 |
| OOP spending as % of the total HIV spending | 0.60 | 0.19 | 0.002 |
| DAHS per total HIV spending ratio | 0.22 | 0.08 | 0.004 |
| Number of nurses per 10K people | -0.57 | 0.18 | 0.002 |
| Density health post per 100K people | 0.11 | 0.06 | 0.082 |

*(Continued)*

**Table 5.** (Continued)

| | | | |
|---|---|---|---|
| GS per total HIV spending ratio | -0.57 | 0.40 | 0.163 |
| External expenditure as % of CHE | -0.08 | 0.08 | 0.287 |
| Sigma | 4.96 | | |

*Notes*: GNI: Gross National Income, CHE: Current Health Expenditure, GDP: Gross Domestic Product, HDI: Human Development Index, OOP: Out-Of-Pocket, DAHS: Development assistance for health spending, GS: Government spending. Coeff.: coefficients, SE: standard errors.

with technical efficiency ranging from 22% to 98%, which suggests that some countries can substantially increase (by up to 78%) levels of output for the same amount of spending.

Countries with higher HIV-burdens did not appear much more likely to have higher spending on HIV (Pearson's correlation coefficient = 0.19). Also, high HIV-burden countries had the highest levels of inputs and output in our analyses which is driving the efficiency scores resulting in a better distribution of the resources (Fig F of Section B in S1 Text). Some of the least efficient countries identified here, such as Guinea-Bissau or the Central African Republic, were also found to be among the least efficient in recent DEA analyses of TB spending [38] and spending for UHC [37]–both of which use the Simar-Wilson approach. Similarly, some of the most efficient countries identified in this analysis, such as Rwanda or Zimbabwe, are also among the most efficient in the analyses of TB and UHC spending. Rwanda and Zimbabwe are also two of the 14 countries that have achieved the target of 73% of PLHIV having suppressed viral loads [5], and Rwanda is among the countries singled out as most efficient by the previous analysis of the technical efficiency of HIV spending in 68 LMICs between 2002–2007 carried out by Zeng and colleagues [22].

Overall, our results therefore highlight that despite improvements over time there is substantial variation in technical efficiency and there is still room to enhance performance–especially in countries with high-HIV burden. By and large, these findings are comparable to those in the existing literature [13, 14, 17, 22, 36]. The key paper by Zeng and colleagues found that the efficiency of HIV spending in 68 LMICs increased from 13.3% to 47.7% between 2002 to 2007 [22]. This is in line with our results, which showed a continued improvement in global technical efficiency between 2010 and 2018 by similar levels. However, there are noteworthy differences, strengths and limitations of our study compared with Zeng and colleagues. First, our sample includes a larger number of countries, years and high-income countries, which were found to be among the most efficient and therefore may have further reduced the estimated efficiency of the lowest performing countries. The smaller sample of countries and years available to Zeng and colleagues at the time may have inflated estimated efficiency. Second, Zeng and colleagues use a traditional two-stage DEA approach, while the double-bootstrap method used in this analysis corrects for bias which reduces estimated levels of efficiency. Third, we only include two of the three outputs used by Zeng and colleagues as voluntary counselling and testing was not considered in this analysis due to data not available.

We find independent variables that consistently affect country efficiency estimates across all models tested. These include Rule of law, HDI, CHE per capita, GNI, OOP spending as % of the GDP, and HIV prevalence, which is in line with previous evidence [13, 22, 36]. However, these associations should not be interpreted as causal. Indeed, CHE per capita is found to have a negative association with efficiency of HIV spending. This is likely reflective of the high performance of low- and low-middle and upper-middle income countries, and large differences in CHE per capita between WB income country groups for similar levels of achieved efficiency

(Fig G of Section B in S1 Text). That said, CHE as a percentage of GDP is positively associated with higher efficiency. This is likely because high and upper-middle income countries invested more as a percentage of their GDP in health than lower- and lower-middle income countries. Another noteworthy negative association is between the number of health posts and average efficiency, which is weak but to our knowledge has not yet been investigated in other similar analyses. Other studies that measured efficiency of HIV spending using cost-effectiveness analyses reported that overall efficiency may depend on the funding available as well as factors such as epidemic response, national targets, societal and development indicators [14, 17, 43]. On the whole, our results indicate that countries with higher HIV prevalence and more nurses are less and more efficient, respectively. In turn, however, the association driven by the health posts might be an indication of high levels of inpatient care and hospitalisation, both of which result in lower efficiency scores. Moreover, the number of nurses was negatively correlated with health posts (Pearson's correlation coefficient = -0.15).

In terms of policy implications, improvements at a country-level may be addressed through different ways of expanding HIV/AIDS services and program coverage. First, countries could increase their budgets towards universal coverage of ART for PLHIV and ARV for pregnant women in need. This is the case in countries with the highest efficiency scores (e.g., Cameroon, Cuba, Dominican Republic, Mozambique, Suriname). They achieved this through the development of national monitoring and evaluation systems, as well as the corresponding local support for resource mobilization and programme implementation to scale-up HIV/AIDS services [44]. Second, countries need to improve the administration of heath resources and routine tracking of technical efficiency. As estimated by the WHO, there is a large degree of inefficiency in the health sector at a global level, which ranges between 20–40% of total health-spending [45]. Our estimates are in line with this and indicate an average inefficiency of 27–31%. Therefore, measures on the capacity of public health systems to deliver good diagnostic, prevention, and treatment, are crucial for a prolonged sustainability. This is especially important for less efficient countries with high HIV-burden that need to address their immediate needs through the re-allocation of their resources (e.g., Democratic Republic of the Congo, Equatorial Guinea, Guinea-Bissau). While efficiency gains might help to expand fiscal and budgetary space for HIV/AIDS, they are insufficient to address the gap between current spending and projected resource needs to achieve the end of the AIDS epidemic by 2030 [46]. Third, improvements in other areas enabling better population health, such as universal healthcare access and stewardship/counselling programs, may enhance technical efficiency by promoting awareness of HIV and equitable access to healthcare [47].

While the sensitivity analysis indicated that findings are robust, some limitations must be considered when interpreting the results. First, we removed a large number of countries (mainly high income) due to substantial missing data in input and output variables. It is possible that including these countries in a future analysis changes our estimates. Second, the DEA method only utilises a single frontier approach (calculated from pooled DMUs) without incorporating a multiple frontier perspective to account for group heterogeneity within DMUs (divergent frontiers). Also, DEA uses an hypothetical comparator rather than an existing (real-life) DMU [48], which might again mask the estimates. However, our analysis is based on publicly available sources and efficiency scores were corrected by accounting for potential biases– initial efficiency scores were overestimated by 1.6% before correction for bias. Also, the DEA approach is more flexible to account for risk biases due to its non-parametric feature. DEA does not require, as deterministic and stochastic parametric methods do, the specification of a functional form for the production frontier [42]. Third, some potential independent variables were not included due to insufficient data, such as information on HIV policy, laws or budgeting processes as well as broader indicators on key characteristics like progress toward

Sustainable Development Goals such as Universal Health Coverage. For instance, the ease of applying Trade-related Aspects of Intellectual Property (TRIPS) flexibilities should be accounted for because both ART and PMTCT programs have a large component of expenditure devoted to commodities, so they are largely constrained by how easy it is to apply TRIPS flexibilities to ensure access to medicines for all the country population [49]. Future iterations of technical efficiency analyses should consider such variables as more data becomes available. Fourth, a measure of quality of care is not included in this analysis, which is a common shortfall of efficiency analyses.

Moving forward, future research may include a larger sample of countries and use additional data as this becomes available on other inputs and outputs such as HIV testing services, and HIV spending by activity or program to obtain more informative estimates. The efficiency scores from this analysis can also be used to undertake a resource needs analysis by decomposing the performance gap into efficiency and resource gaps, as done by the follow-up study to the HIV efficiency analysis previously carried out by Zeng and colleagues [36]. In addition, future analyses should aim to analyse how efficient countries are when considering TB and HIV spending and outcomes combined–the importance of which is clear in emerging literature [50]. For instance, TB patients tested positive for HIV (and-or receiving ART), and joint attributed mortality, are also a concern in high HIV-burden countries (e.g., Southern and Central African countries) which will likely affect their technical efficiency. Nonetheless, the results of this study can help governments and donors by providing a benchmark to facilitate improvements in the efficiency of converting HIV spending into service coverage to accelerate progress towards 2030 targets for eradication [5]. However, even some of the most efficient countries in this analysis such as Romania and Ethiopia have not made sufficient progress toward global HIV targets. Combined with the negative impact of the COVID-19 pandemic on progress toward global HIV targets, this study highlights the need for additional investment to develop new approaches in addressing HIV programs as well as broader investment in health and social protection. Optimal and timely investment and distribution of goods and services for the general population, but specifically those at risk, are crucial for better epidemiological and financial sustainability.

## Conclusion

The present study used the double-bootstrap DEA to examine the technical efficiency of national HIV/AIDS spending and independent variables associated with efficiency in 78, mostly low- and-middle income, countries between 2010 and 2018. Our findings suggested that, on average, outputs could have increased by 18.2% in 2018 for the same amount of national HIV/AIDS spending. Efficiency scores have varied by income group, and geographical region, but exhibit sustained improvement over the years. Rule of Law, GNI per capita, reduced out-of-pocket expenditure as a % of the total HIV spending, and HDI were associated with technical efficiency. Our sensitivity analyses showed that our predicted efficiency scores did not vary ($< 1$ percentage point), which suggests our results are robust. Given that even the most efficient countries did not meet global 2020 HIV targets, our study supports the WHO call for additional investment in HIV/AIDS prevention and control to meet the 2030 global targets for viral load suppression and eradication of the AIDS epidemic.

## Supporting information

**S1 Text.**
(DOCX)

**S1 Data.**
(XLSX)

**S2 Data.**
(XLSX)

## Author Contributions

**Conceptualization:** Kasim Allel, Gerard Joseph Abou Jaoude, Hassan Haghparast-Bidgoli.

**Data curation:** Kasim Allel, Gerard Joseph Abou Jaoude.

**Formal analysis:** Kasim Allel, Gerard Joseph Abou Jaoude.

**Investigation:** Charles Birungi, Tom Palmer, Hassan Haghparast-Bidgoli.

**Methodology:** Kasim Allel, Gerard Joseph Abou Jaoude, Charles Birungi, Tom Palmer, Hassan Haghparast-Bidgoli.

**Project administration:** Kasim Allel, Gerard Joseph Abou Jaoude.

**Software:** Kasim Allel.

**Supervision:** Kasim Allel, Gerard Joseph Abou Jaoude, Charles Birungi, Jolene Skordis, Hassan Haghparast-Bidgoli.

**Validation:** Kasim Allel, Gerard Joseph Abou Jaoude, Tom Palmer, Hassan Haghparast-Bidgoli.

**Visualization:** Kasim Allel.

**Writing – original draft:** Kasim Allel, Gerard Joseph Abou Jaoude.

**Writing – review & editing:** Kasim Allel, Gerard Joseph Abou Jaoude, Charles Birungi, Tom Palmer, Jolene Skordis, Hassan Haghparast-Bidgoli.

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
