## [Decision Letter · Decision Letter 0]

1 Feb 2022

PGPH-D-21-00738

Technical efficiency of national HIV/AIDS spending in 78 low- and middle-income countries between 2010 and 2018: a data envelopment analysis

Dear Dr. Abou Jaoude,

Thank you for submitting your manuscript to PLOS Global Public Health. After careful consideration, we feel that it has merit but does not fully meet PLOS Global Public Health’s publication criteria as it currently stands. Therefore, we invite you to submit a revised version of the manuscript that addresses the points raised during the review process.

COMMENTS FROM THE EDITOR:

First of all, my apologies for the delay in processing your manuscript. A couple of reviewers reneged on their initial willingness to review the manuscript, which explain part of the delay.

I have now read the manuscript alongside two great reviews. I agree with both reviewers that the manuscript is done nicely but that it could benefit from a number of changes. Reviewer 1 has also put a number of additional comments in the attached manuscript. Please address all comments from the reviewers. It is fine, however, if you decide not to worry about normality that one of the reviewers points out.

When you submit a revised piece, I request you to address the following additional comments as well. Let us know if additional clarity is needed on my or the reviewers’ comments.

MAJOR: It was not clear to me why many of the independent variables you have used would affect efficiency. You have avoided making causal statements, which is appreciated. However, in Table 2, the discussion of why ‘antenatal care coverage’ (for example) would be associated with efficiency is insufficient. Some of the other ‘independent variables’ might already be highly correlated with factors that feed into the efficiency measure (for example, “OOP spending as a % of total HIV spending” and “national spending on HIV”). In that sense, even the variables used in the sensitivity analysis—e.g., number of nurses—might already be reflecting what feeds into the efficiency measure. Methodological issues like these become more visible here because there are some striking findings—e.g., negative relationship between rule of law and technical efficiency—which are hard to explain. Unless I am missing something important, my suggestion is to start from a really parsimonious model where you put in factors that should really matter for efficiency based on findings from previous studies, and build from there.  I should note that the documentation of changes in technical efficiency over time and across geographies is the most important contribution of this paper anyway. [Minor: for Tables 4 and 5, consider condensing them. No need to report 95% CI, SE, Z and p-values. Perhaps coefs and SEs, with *, **, and *** to denote conventional cutoffs for statistical significance will suffice?] MAJOR: Many readers of this journal will not be familiar with DEA. Therefore, please make sure that you explain DEA in simpler terms. This is particularly important because the (efficiency in HIV/AIDS spending) will be of high interest to researchers and policymakers from around the world. The current version seems to assume a lot on the part of the reader’s knowledge. For more technical summary, you can refer the reader to previous studies—as you have done now. MINOR: Some notes on the discussion.a. You have spent half-a-paragraph discussing potential reason for the difference of a 0.2 percentage point between your findings and those of Zeng et al. That seems redundant.b. Some of the policy implications are not directly based on your findings (“national efforts to improve SDGs..”). Consider removing them.c. Among the limitations, that the analysis relies on observed data and the data may not be reliable is a problem with any study. Explain why you think the data might be unreliable here, and on which direction your findings would be biased because of that. Otherwise, fine to remove it. Generally, consider organizing the limitations better (e.g., the one about measurement error, lack of info on some variables can be lumped into an umbrella limitation about the lack of causality).  d. The sentence about “multiple frontier approach” is unclear.4. MINOR: A more general comment. You may wish to get the paper edited by a professional editor as there are several places in the text where writing can be improved. To point out a few issues, some of the paragraphs are very long, I noticed a few sentences repeated, and there are a good number of redundant sentences (e.g., “In healthcare, economics is typically applied to measure allocative or technical efficiency”, “Data on quality of care would be beneficial both to this study and other technical and allocative efficiency studies”).  

We look forward to receiving your revised manuscript.

Kind regards,

Yubraj Acharya, Ph.D.

Academic Editor

Journal Requirements:

1. Please amend your Financial Disclosure statement. If you did not receive any funding for this study, please simply state: “The authors received no specific funding for this work.”

2. Please update your Competing Interests statement. If you have no competing interests to declare, please state: “The authors have declared that no competing interests exist.”

3. Please note that your Data Availability Statement is currently missing direct links to access each database. If your manuscript is accepted for publication, you will be asked to provide these details on a very short timeline. We therefore suggest that you provide this information now, though we will not hold up the peer review process if you are unable.

4. HIV_DEA_28Sep21_SUPPLEMENTARY.docx is over our file size limit of 20MB. This limit is in place for the convenience of reviewers, editors and readers. Please adapt these files so that the file size is below 20MB.

You may find it helpful to consult our guidelines on compressing figures here: https://journals.plos.org/globalpublichealth/s/figures

5. Please provide separate figure files in .tif or .eps format only and ensure that all files are under our size limit of 20MB.

For more information about how to convert your figure files please see our guidelines: 

6. Please provide us with a direct link to the base layer of the map used in Figure 2 and ensure this location is also included in the figure legend. 

Please note that, because all PLOS articles are published under a CC BY license (creativecommons.org/licenses/by/4.0/), we cannot publish proprietary maps such as Google Maps, Mapquest or other copyrighted maps. If your map was obtained from a copyrighted source please amend the figure so that the base map used is from an openly available source.

Please note that only the following CC BY licences are compatible with PLOS licence: CC BY 4.0, CC BY 2.0  and CC BY 3.0, meanwhile such licences as CC BY-ND 3.0 and others are not compatible due to additional restrictions. If you are unsure whether you can use a map or not, please do reach out and we will be able to help you. 

The following websites are good examples of where you can source open access or public domain maps:

Reviewers' comments:

Reviewer's Responses to Questions

**Comments to the Author**

1. Does this manuscript meet PLOS Global Public Health’s publication criteria? Is the manuscript technically sound, and do the data support the conclusions? The manuscript must describe methodologically and ethically rigorous research with conclusions that are appropriately drawn based on the data presented.

Reviewer #1: Yes

Reviewer #2: Yes

2. Has the statistical analysis been performed appropriately and rigorously?

Reviewer #1: Yes

Reviewer #2: Yes

3. Have the authors made all data underlying the findings in their manuscript fully available (please refer to the Data Availability Statement at the start of the manuscript PDF file)?

Reviewer #1: No

Reviewer #2: Yes

4. Is the manuscript presented in an intelligible fashion and written in standard English?

Reviewer #1: Yes

Reviewer #2: Yes

5. Review Comments to the Author

Reviewer #1: This secondary data analysis to assess the technical efficiency of HIV/AIDS spending in 78 countries is a well written paper. The Data Enveloping Analysis deployed for the economic analysis is both adjudged sensitive and novel.

The following are the specific observation made on the manuscripts;

Comment 1. The authors should check for some general comments in the body of the manuscript

Comment 2. A brief description of importance of economic efficiency, types of measurement and DEA should be included in the introduction section.

Comment 3. The second research question is similar to second part of the first research question. This should be rephrased

Comment 4. The first two paragraphs of the methods section are placed wrongly. The long description of the DEA should come partly in the introduction while the second paragraph should be in the description of the methods of data analysis

Comment 5: The authors should include the operational definitions of how countries included in the study were classified according to WHO and/or WB etc. This should be in the subsection "Sample and main data sources"

Comment 6. There is confusion of the number of countries included in this study. The 81 countries written in the methods is in contrast to what was declared in Figure 1, Table 1, summary section and in the results. This should be corrected

Comment 7. There are some discrepancies between the title of the study and the conclusion of this study. While the title of the study was specific to LMICS, the results and conclusion generalize on World Bank classification of countries by HDI. The authors should adjust the title to reflect this.

Comment 8. The authors should explain in details the unexpected inverse relationship between rule of law and HDI to HIV technical efficiency more.

Comment 9. Table 5: These tables are not properly labeled. Each of the tables should caption what form of sensitivity analysis was carried out

Comment 10. Can the authors explain why the exclusion of voluntary counseling and testing could have led to reduction in the estimated output

Comment 11. Could multicolinearity between the independent variables be a major consideration by authors in reporting and interpreting the result? The authors should explain in detailed how this was addressed in methods and in the discussion sections

Comment 12. The supplementary data should be included

Reviewer #2: This is a good paper on technical efficiency on HIV/AIDS spending in 78 low- and middle-income countries. The paper is well-written, and methodologically sound. The result section provides interesting findings and leads to important observations that are well supported and discussed in the discussion section. Policy implications and limitations are described as well.

In conclusion: great work!

I have a few comments for the authors.

Under “Input and output variables”, the authors state that “To generate the DEA input, annual national spending on HIV was divided by the annual number of people notified with HIV”, does this mean this only included new cases only who were notified of their HIV status? Spending would include both new+existing cases.

In Table 1, the authors could add a column of percentage out of the total in each category to show the coverage across regions. E.g African region 36/total countries in African region.

To handle missing data the authors used mean imputation and the last observation carried forward methods. They also used multiple imputation. Authors could provide an equation for the regression model used for the multiple imputations. Could the authors also expound on any diagnostics to assess whether the imputations were robust. There are several methods of doing this for instance compared imputed with observed values, checking model convergence etc. Authors could refer https://ete-online.biomedcentral.com/articles/10.1186/s12982-017-0062-6

Authors could also carry out complete case analysis (when all variables are not imputed) and compared results with those from imputed data as sensitivity analysis

Were the variables assessed for normality? The distribution of non-normal variables could be presented using median and interquartile range in Table 3.It is possible some variables would vary among different countries and hence the skew the distribution of such variables.

6. PLOS authors have the option to publish the peer review history of their article (what does this mean?). If published, this will include your full peer review and any attached files.

**Do you want your identity to be public for this peer review?** For information about this choice, including consent withdrawal, please see our Privacy Policy.

Reviewer #1: No

Reviewer #2: **Yes: **Steven Wambua

---

## [Decision Letter · Decision Letter 1]

27 Jun 2022

Technical efficiency of national HIV/AIDS spending in 78 countries between 2010 and 2018: a data envelopment analysis

PGPH-D-21-00738R1

Dear Mr Abou Jaoude,

I hope you are doing well. We are pleased to inform you that your manuscript 'Technical efficiency of national HIV/AIDS spending in 78 countries between 2010 and 2018: a data envelopment analysis' has been provisionally accepted for publication in PLOS Global Public Health.

Best regards,

Yubraj Acharya, Ph.D.

Academic Editor

Reviewer Comments (if any, and for reference):

Reviewer's Responses to Questions

**Comments to the Author**

1. If the authors have adequately addressed your comments raised in a previous round of review and you feel that this manuscript is now acceptable for publication, you may indicate that here to bypass the “Comments to the Author” section, enter your conflict of interest statement in the “Confidential to Editor” section, and submit your "Accept" recommendation.

Reviewer #2: All comments have been addressed

2. Does this manuscript meet PLOS Global Public Health’s publication criteria? Is the manuscript technically sound, and do the data support the conclusions? The manuscript must describe methodologically and ethically rigorous research with conclusions that are appropriately drawn based on the data presented.

Reviewer #2: Yes

3. Has the statistical analysis been performed appropriately and rigorously?

Reviewer #2: Yes

4. Have the authors made all data underlying the findings in their manuscript fully available (please refer to the Data Availability Statement at the start of the manuscript PDF file)?

Reviewer #2: Yes

5. Is the manuscript presented in an intelligible fashion and written in standard English?

Reviewer #2: Yes

6. Review Comments to the Author

Reviewer #2: I am satisfied with all the responses to the comments I had raised in the first review.

7. PLOS authors have the option to publish the peer review history of their article (what does this mean?). If published, this will include your full peer review and any attached files.

**Do you want your identity to be public for this peer review?** For information about this choice, including consent withdrawal, please see our Privacy Policy.

Reviewer #2: **Yes: **Steven Wambua
